# Sonographic, Demographic, and Clinical Characteristics of Pre- and Postmenopausal Women with Endometrial Cancer; Results from a Post Hoc Analysis of the IETA4 (International Endometrial Tumor Analysis) Multicenter Cohort

**DOI:** 10.3390/diagnostics14010001

**Published:** 2023-12-19

**Authors:** Rasmus W. Green, Daniela Fischerová, Antonia C. Testa, Dorella Franchi, Filip Frühauf, Pelle G. Lindqvist, Alessia di Legge, David Cibula, Robert Fruscio, Lucia A. Haak, Gina Opolskiene, Ailyn M. Vidal Urbinati, Dirk Timmerman, Tom Bourne, Thierry van den Bosch, Elisabeth Epstein

**Affiliations:** 1Department of Clinical Science and Education, Karolinska Institute, Södersjukhuset, Sjukhusbacken 10, 118 83 Stockholm, Sweden; pelle.lindqvist@ki.se (P.G.L.);; 2Department of Gynaecology, Obstetrics and Neonatology, General University Hospital and First Faculty of Medicine, Charles University, Apolinářská 18, 128 51 Prague, Czech Republic; daniela.fischerova@seznam.cz (D.F.); fruhauffilip@centrum.cz (F.F.); dc@davidcibula.cz (D.C.); 3Department of Women and Child Health, Division of Gynecologic Oncology, Fondazione Policlinico Universitario A. Gemelli IRCCS, 00168 Rome, Italy; testa.antonia@gmail.com; 4Department of Life Science and Public Health, Catholic University of Sacred Heart Largo Agostino Gemelli, 00168 Rome, Italy; 5Department of Gynecological Oncology, European Institute of Oncology (IEO) IRCCS, 20141 Milan, Italy; dorella.franchi@ieo.it (D.F.); ailyn.vidalurbinati@ieo.it (A.M.V.U.); 6Department of Obstetrics and Gynecology, Södersjukhuset, 118 83 Stockholm, Sweden; 7Department of Obstetrics and Gynecology, Catholic University of the Sacred Heart, 00168 Rome, Italy; alessiadilegge0@gmail.com; 8UO Gynecology, Department of Medicine and Surgery, IRCCS San Gerardo, University of Milan Bicocca, 20126 Milan, Italy; robert.fruscio@unimib.it; 9Institute for the Care of Mother and Child, Prague and Third Faculty of Medicine, Charles University, 147 00 Prague, Czech Republic; 10Center of Obstetrics and Gynecology, Faculty of Medicine, Vilnius University Hospital, 08661 Vilnius, Lithuania; gina.opolskiene@santa.lt; 11Department of Development and Regeneration, KU Leuven, 3000 Leuven, Belgium; dirk.timmerman@uzleuven.be (D.T.); thierryvandenbosch1901@gmail.com (T.v.d.B.); 12Department of Obstetrics and Gynecology, University Hospital Leuven, 3000 Leuven, Belgium; 13Department of Obstetrics and Gyneacology, Queen Charlotte’s and Chelsea Hospital, Imperial College London, London W12 0HS, UK

**Keywords:** endometrial neoplasms, premenopause, perimenopause, postmenopause, biometry, ultrasonography, risk factors, lifestyle, exploratory research

## Abstract

In this study, we conducted a comparative analysis of demographic, histopathological, and sonographic characteristics between pre- and postmenopausal women diagnosed with endometrial cancer, while also examining sonographic and anthropometric features in ‘low’ and ‘intermediate/high-risk’ cases, stratified by menopausal status. Our analysis, based on data from the International Endometrial Tumor Analysis (IETA) 4 cohort comprising 1538 women (161 premenopausal, 1377 postmenopausal) with biopsy-confirmed endometrial cancer, revealed that premenopausal women, compared to their postmenopausal counterparts, exhibited lower parity (median 1, IQR 0–2 vs. 1, IQR 1–2, *p* = 0.001), a higher family history of colon cancer (16% vs. 7%, *p* = 0.001), and smaller waist circumferences (median 92 cm, IQR 82–108 cm vs. 98 cm, IQR 87–112 cm, *p* = 0.002). Premenopausal women more often had a regular endometrial–myometrial border (39% vs. 23%, *p* < 0.001), a visible endometrial midline (23% vs. 11%, *p* < 0.001), and undefined tumor (73% vs. 84%, *p =* 0.001). Notably, despite experiencing a longer duration of abnormal uterine bleeding (median 5 months, IQR 3–12 vs. 3 months, 2–6, *p* < 0.001), premenopausal women more often had ‘low’ risk disease (78% vs. 46%, *p* < 0.001). Among sonographic and anthropometric features, only an irregular endometrial–myometrial border was associated with ‘intermediate/high’ risk in premenopausal women. Conversely, in postmenopausal women, multiple features correlated with ‘intermediate/high’ risk disease. Our findings emphasize the importance of considering menopausal status when evaluating sonographic features in women with endometrial cancer.

## 1. Introduction

Endometrial cancer (EC) ranks as the sixth most prevalent global cancer in women, comprising 4.8% of all female malignancies [1]. Its incidence rate in Europe and North America averages 13–19 cases per 100,000 women annually [2]. Only 2.8–6.5% of new EC diagnoses occur in women under 45, as reported by national statistics from the US [3], the UK [4], and Sweden [5]. Among premenopausal women with abnormal uterine bleeding, the EC risk stands at a mere 0.33% [6], contrasting with a substantially higher 9% (95% confidence interval: 8–11%) risk observed in postmenopausal bleeding cases [7]. Established risk factors encompass obesity, metabolic syndrome [8], early menarche, and late menopause [9]. In premenopausal women, obesity [10,11] and nulliparity [10,12] have emerged as confirmed risk factors. Younger patients typically present with earlier-stage, lower-grade disease and have significantly better disease-specific survival rates [13,14].

Expert ultrasound or MRI evaluations are recommended for assessing local tumor extension and identifying women at intermediate/high risk [15]. The combination of imaging and pre-operative biopsy histological grading achieves approximately 80% diagnostic accuracy in identifying high-risk cases [16]. Sonographic characteristics correlate with histological risk groups, underscoring an important role in presurgical triage [17]. However, despite the importance of menopausal status in shaping histological risk features and potentially sonographic findings, a noteworthy gap in the literature exists regarding high-quality studies comparing sonographic characteristics between pre- and postmenopausal EC patients.

To address this gap, we conducted an analysis utilizing data from the prospective International Endometrial Tumor Analysis (IETA) 4 cohort, comprising 1538 women, exploring multiple sonographic, demographic, and sonographic characteristics. While previous publications have presented results from this cohort [16,17,18,19], none have explored these findings while stratifying for menopausal status. Since these data have been analyzed previously, our study is post hoc and exploratory in nature. Therefore, our study (1) seeks to explore disparities in demographic, anthropometric, lifestyle, clinical, and (2) sonographic characteristics between pre- and postmenopausal women with EC and (3) assess if sonographic and anthropometric characteristics relate to risk classification differentially between pre- and postmenopausal women, to generate hypotheses for future confirmatory investigations.

## 2. Materials and Methods

### 2.1. Patients

In our investigation, we used the prospective IETA4 multicenter cohort, which encompassed 1538 women diagnosed with biopsy-confirmed endometrial cancer. Women were recruited from 17 gynecologic oncology centers consecutively between 1 January 2011 and 31 December 2015. Expert ultrasound examiners used high-end ultrasound systems and adhered to the IETA4 study protocol for scanning [17]. The examination procedure involved transvaginal scans, conducted with an empty bladder in the lithotomy position, complemented by transabdominal scans when necessary. The categorization of endometrial grayscale morphology and power Doppler findings followed the established IETA terminology [20]. Data are extensive, encompassing medical, reproductive, and gynecological history, family history, presenting symptoms, lifestyle factors, demographic characteristics, anthropometric measurements, and sonographic findings, and were recorded in an online database software designed for the IETA collaboration (the Clinical Data Miner, v. 0.0.1 at the start of data entry, v. 1.0 at the time of data withdrawal; https://cdm.esat.kuleuven.be/CDM/), accessed on 28 December 2017 [21]. Data entry was carried out promptly after history taking and examination, with no provision for saving incomplete or altering submitted data. To ensure consistency among examiners, the Clinical Data Miner featured pictograms for each finding for direct comparison. Outcome data, including histology, grade, and International Federation of Gynecology and Obstetrics (FIGO) surgical stage, were submitted following hysterectomy. Pathological assessments were conducted by experienced gynecologic oncology pathologists at each center, adhering to FIGO 2009 staging criteria based on surgical observations [22]. Only malignant epithelial tumors, i.e., endometrial carcinomas (endometrioid, mucinous, serous, clear cell, mixed, and undifferentiated carcinomas), and malignant mixed epithelial and mesenchymal tumors, i.e., carcinosarcomas (malignant mixed Müllerian tumor), were included [23]. A comprehensive description of this database and examination protocol is available in prior publications [17].

### 2.2. Variables

Participants in this study were categorized by menopausal status, with postmenopausal defined as a minimum of 12 consecutive months of amenorrhea. Comprehensive demographic and clinical data were collected through self-reported questionnaires. These data encompassed age, parity, the presence of abnormal bleeding as a presenting symptom (distinguished as ‘yes’ or ‘no’), the duration of abnormal bleeding if applicable (measured in months), family history of cancer (including ovarian, breast, colon, or other cancers among first-degree relatives), co-morbidities (specifically, diabetes mellitus type 2 and hypertension), use of hormonal treatment (for any indication), smoking status (‘never’, ‘former’, ‘present’), levels of physical activity (classified into five categories ranging from ‘never’ to ‘strenuous activity ≥ 2 times/week’), alcohol consumption (divided into four levels from ‘never’ to ‘>12 units/week’), and bra cup size (US system, from AA to I). Lynch syndrome data were not included in the dataset.

Anthropometric measurements, such as weight (in kilograms), height (in centimeters), body constitution (subjectively categorized as ‘lean’, ‘abdominal adiposity’, or ‘female adiposity’), and waist circumference (in centimeters), were directly assessed by the examining physician or an assisting nurse. Body mass index (BMI) was calculated from weight and height data and dichotomized using a threshold of ≥30 kg/m², while waist circumference was dichotomized at ≥88 cm [24].

Histopathological measures relied on hysterectomy findings and included FIGO surgical stage (‘IA’ or ‘≥IB’), histological type (‘endometrioid grade 1 + 2’ or ‘endometrioid grade 3 + non-endometrioid’). Due to the absence of data on lymphovascular space invasion, a modified postoperative risk grouping, aligned with the ESMO-ESGO-ESTRO consensus, was applied, dividing cases into ‘low risk’ and ‘intermediate/high risk’. Specifically, ‘low risk’ encompassed endometrioid histology of grade 1 or 2 with FIGO stage IA, while all other cases, including grade 3 endometrioid, non-endometrioid histology, and FIGO stage ≥IB, were categorized as ‘intermediate/high risk.’ Consequently, the ‘intermediate’ and ‘high’ risk groups were combined for the purposes of our analysis. Sonographic characteristics were classified in accordance with the IETA consensus criteria [20] and grouped based on the initial findings of the IETA4 study [17]. These characteristics included endometrial measurability, tumor definition, presence of fibroids, suspicion of adenomyosis, appearance of the endometrial–myometrial border, endometrial echogenicity and uniformity, presence of a bright edge sign, assessment of the endometrial midline, color score, vascular pattern, endometrial thickness, and tumor volume.

### 2.3. Statistical Analyses

Continuous variables are presented with medians and interquartile ranges and categorical variables by frequencies and percentages. Comparisons between groups were conducted using the Mann–Whitney *U* test for continuous variables and Fisher’s exact test for categorical variables. With multiple hypotheses testing in an exploratory approach, the risk of falsely rejecting a null hypothesis is high. Therefore, we used the Benjamini–Yekutieli procedure with its modification for arbitrary dependency between *p*-values [25] on all hypothesis testing. Stata 16.1/IC (StataCorp, College Station, TX, USA) was used for all calculations, with the user-written command multproc for the Benjamini–Yekutieli procedure [26].

## 3. Results

### 3.1. Demographic and Anthropometric Characteristics

Among the 1538 patients in our study, 161 (10%) were premenopausal, while 1377 (90%) were postmenopausal. Demographic characteristics, lifestyle factors, anthropometric measurements, and clinical data comparing the two groups are detailed in Table 1. Notably, premenopausal women exhibited lower parity (median 1, IQR 0–2 vs. median 2, IQR 1–2, *p* < 0.001) and a higher incidence of familial colon cancer history (16% vs. 7%, *p* = 0.001). Although approximately two-fifths of both pre- and postmenopausal women were obese, premenopausal individuals had a smaller waist circumference (median 92, IQR 82–108 vs. median 98, IQR 87–112, *p* = 0.002). Abnormal uterine bleeding was as common in both groups, with premenopausal women reporting a longer duration of symptoms (median 5 months, IQR 3–12 vs. median 3 months, IQR 2–6, *p* < 0.001).

### 3.2. Histopathological Characteristics

Table 2 outlines the distribution of FIGO surgical stage, histology, and ESMO-ESGO-ESTRO risk groups among pre- and postmenopausal women. Notably, premenopausal women more frequently presented with FIGO stage IA tumors (84% vs. 58%, *p* < 0.001), endometrioid histology (98% vs. 85%, *p* < 0.001), lower-grade tumors (*p* = 0.0012, Mann–Whitney *U* test for difference in rank), and a higher prevalence of ‘low’ ESMO-ESGO-ESTRO risk group cases (78% vs. 46%, *p* < 0.001) compared to postmenopausal women. Due to the low frequency of non-endometrioid tumors, differences in histologic subtypes among these tumors could not be statistically assessed.

### 3.3. Sonographic Characteristics

Sonographic findings, as presented in Table 3, reveal notable distinctions between pre- and postmenopausal women. Premenopausal women had a tumor that was defined on ultrasound less often (73% vs. 84%, *p* = 0.001), a higher suspicion of adenomyosis (16% vs. 7%, *p* = 0.001), more frequently a regular endometrial–myometrial border (39% vs. 23%, *p* < 0.001), and more often a visible endometrial midline (23% vs. 11%, *p* < 0.001) compared to their postmenopausal counterparts. However, no significant differences emerged in endometrial measurability, fibroid prevalence, endometrial echogenicity, bright edge sign, color score, vascular pattern, endometrial thickness, or tumor volume between the two groups.

### 3.4. Endometrial Thickness in Relation to Anthropometric Characteristics

In premenopausal women, endometrial thickness did not significantly differ between those with BMI < 30 and BMI ≥ 30 (median 14 mm, IQR 10–23 vs. median 15 mm, IQR 10–22, *p* = 0.605) or between those with waist circumference <88 cm and ≥88 cm (median 14 mm, IQR 9–20 vs. median 15 mm, IQR 10–23.5, *p* = 0.219). In postmenopausal women, a subtle difference in endometrial thickness emerged between those with BMI < 30 and BMI ≥ 30 (median 16 mm, IQR 9–26 vs. median 18 mm, IQR 11–26, *p* = 0.036) and between those with waist circumference <88 cm and ≥88 cm (median 15 mm, IQR 8–24 vs. median 17 mm, IQR 10–26, *p* = 0.005). However, it is noteworthy that these differences, while statistically significant at the 0.05 level, did not maintain significance following correction by using the Benjamini–Yekutieli procedure (Figure 1 and Figure 2)

### 3.5. Sonographic and Anthropometric Characteristics in Relation to Risk Group

Table 4 provides a comprehensive view of the distribution of sonographic and anthropometric characteristics by ESMO-ESGO-ESTRO risk groups, stratified by menopausal status. Among premenopausal women, the sole sonographic characteristic associated with ‘intermediate/high’ risk was a non-regular endometrial–myometrial border (91% among ‘intermediate/high’ risk vs. 54% among ‘low’ risk, *p* < 0.001). In contrast, among postmenopausal women, several sonographic features, including a defined tumor, a non-regular endometrial–myometrial border, a non-uniform echogenicity, a ‘moderate-abundant’ color score, and a multiple, multifocal vascular pattern, all correlated with ‘intermediate/high’ risk. Both BMI ≥ 30 and waist circumference ≥88 cm were associated with the ‘low’ ESMO-ESGO-ESTRO risk group among postmenopausal women.

### 3.6. Results after Controlling the False Discovery Rate

Overall, our study entailed 53 hypothesis tests, resulting in the initial rejection of the null hypothesis in 24 tests at a significance level of *p* < 0.05. However, following the application of the Benjamini–Yekutieli procedure, the corrected overall critical *p*-value was adjusted to 0.0041. Under this more conservative threshold, 20 hypotheses were validated as true positive findings, while four were identified as false rejections of the null hypothesis, denoting false positives. These four false positives pertained to body constitution, bright edge sign, and the differences in endometrial thickness between the BMI and waist circumference groups among postmenopausal women (Appendix A).

## 4. Discussion

Our study is exploratory in nature and thus only ever hypothesis-generating [27]. Still, some valuable insights into the differences in sonographic, anthropometric, and clinical features between pre- and postmenopausal women with endometrial cancer can be found. Most notably, we saw a differential association between ultrasound findings and risk group designation. In premenopausal women, we observed a less frequent definition of tumors and a more regular endometrial–myometrial border, with only the latter being associated with risk group. Conversely, in postmenopausal women, several sonographic features, including defined tumors, irregular endometrial–myometrial borders, non-uniform echogenicity, elevated color scores, and a multiple, multifocal vessel pattern, correlated with ‘intermediate/high’ risk. Furthermore, a higher BMI and larger waist circumference were linked to low-risk disease in postmenopausal women but without such association in premenopausal women.

Our data revealed that more premenopausal women exhibited no evidence of disease at hysterectomy (6.2% vs. 1.4%), potentially explaining the lower prevalence of defined tumors on ultrasound, as these tumors were likely resected in toto during hysteroscopic resection. This observation aligns with the higher occurrence of FIGO stage IA cancers among premenopausal women, which typically entail only superficial myometrial invasion. Given the rarity of endometrial cancer among premenopausal women with abnormal uterine bleeding (about 0.33% [6]), this study could not have been performed on patients prior to endometrial sampling.

While certain variations in specific sonographic characteristics, such as a regular endometrial–myometrial border and a visible endometrial midline, are expected due to the higher prevalence of low-risk cancer among premenopausal women, our findings go beyond this expectation. We discovered that sonographic features were not solely dependent on ESMO-ESGO-ESTRO risk groups, as one might anticipate. Had that been the case, we would have observed a higher prevalence of uniform echogenicity, lower color scores, a thinner endometrium, and a lower prevalence of multiple, multifocal vessel patterns in premenopausal women, as these features are common among low-risk cases [17]. In premenopausal women, physiological secretory changes related to fertility can influence endometrial vascularity and thickness, potentially leading to misinterpretation as high-risk sonographic features.

In asymptomatic postmenopausal women, the mean endometrial thickness is around 3 mm [28,29], while in premenopausal women, it ranges from approximately 5 mm at its thinnest to about 11 mm at its thickest, depending on the menstrual cycle’s timing [30]. Although we noted a non-significant difference in median endometrial thickness between the groups (14 mm vs. 17 mm, *p* = 0.13), it is important to highlight that postmenopausal women exhibited more deviation from the normal range (six times thicker vs. two times thicker). Surprisingly, endometrial thickness did not show significant associations with BMI or waist circumference in either group after applying the Benjamini–Yekutieli correction, as would be expected with higher endogenous estrogen levels due to obesity.

Our study findings are in line with previous research indicating that obesity is linked to lower-stage, lower-grade, and thus overall low-risk endometrial cancer [31,32,33]. Yet, disease-specific survival is unaffected by BMI [34]. Our study further revealed that a higher BMI (≥30) and a greater waist circumference (≥88 cm) were specifically associated with low-risk disease among postmenopausal women. In our premenopausal group, 66.5% were overweight or obese, compared to 30.9% among 18- to 44-year-olds in the European Union according to official statistics [35].

While it is expected that premenopausal women report longer symptom durations due to the commonality of abnormal uterine bleeding in fertile women and the rarity of EC among them, our study confirms that these women still often present with low-risk tumors. This suggests that even subnormal levels of circulating progesterone [36] in premenopausal women may offer some protective effects against high-risk EC or that genetic random mutations driving high-risk EC have not had time to accumulate in younger women. Thus, our findings underscore that non-endometrioid pathologies are almost exclusively found in postmenopausal women.

This study is one of the largest multicenter prospective cohort studies, enhancing the generalizability of our results. The dedicated data capture software employed likely improved data quality. It is important to note that our power analysis was not initially designed for comparing pre- and postmenopausal women, and this study is exploratory in nature. While our study included only 161 premenopausal women, making it a potential limitation, it remains one of the largest prospective studies integrating detailed sonographic, demographic, and anthropometric data. Since only 35 premenopausal women had intermediate/high-risk EC, it was not possible to perform multivariable regression analysis to adjust for confounding factors between age, menopausal status, risk classification, demographic factors, and sonographic findings. Collinearity between these factors is likely. The overrepresentation of premenopausal cases in our database is attributed to the tertiary nature of the participating centers, where some low-risk cases may have been treated at secondary centers. Importantly, this sampling bias does not skew estimates in group comparisons.

Addressing concerns about multiple exploratory comparisons and the risk of false-positive findings, we mitigated this by implementing the Benjamini–Yekutieli procedure. This approach adjusts the *p*-value threshold to necessitate larger differences for statistical significance, reducing the likelihood of spurious findings [37]. Nonetheless, it is essential to acknowledge that the absence of detected differences does not definitively rule out the existence of differences, and conversely, detected differences can sometimes be spurious. Our findings, where 20 of 24 rejected null hypotheses were retained after correction to a *p*-value threshold of 0.0041, indicate that pre- and postmenopausal EC are distinct entities. We acknowledge that the use of *p*-values, or rather, null hypothesis significance testing (NHST), in an exploratory setting is heavily debated within the statistical community [38,39,40,41] and therefore chose a middle ground by conducting NHST but then controlling the false discovery rate with the use of the Benjamini–Yekutieli procedure.

While our study offers hypothesis-generating insights, these findings must be replicated in future studies before their incorporation into clinical practice. To validate our most clinically relevant discoveries regarding sonographic findings in association with risk group among premenopausal women, a study with as few as 36 low- and intermediate/high-risk premenopausal cases (for a 99% confidence level and 90% power) is required to verify a different proportion of regular to non-regular endometrial–myometrial border. However, a sample of at least 229 premenopausal women is necessary to detect differences in color score and vascular pattern between ‘low’ and ‘intermediate/high’ risk cases at a 97.5% confidence level if a Bonferroni adjustment is applied.

## 5. Conclusions

In premenopausal women with endometrial cancer, only an irregular endometrial–myometrial border is associated with ‘intermediate/high risk’ while vascularity and endometrial echogenicity should be interpreted cautiously since these features likely depend on physiological processes. Finding high-risk features among premenopausal women is difficult considering almost all premenopausal women have low-risk disease, despite reporting longer duration of abnormal uterine bleeding prior to diagnosis.

## Figures and Tables

**Figure 1 diagnostics-14-00001-f001:**
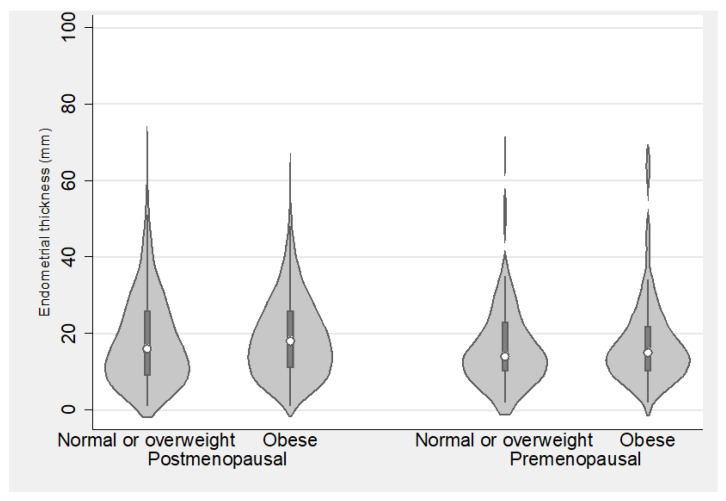
Endometrial thickness among normal/overweight and obese women, stratified by menopausal status.

**Figure 2 diagnostics-14-00001-f002:**
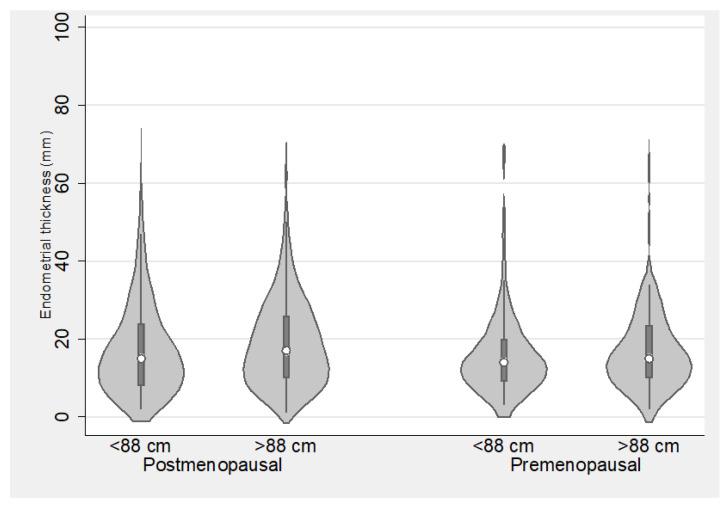
Endometrial thickness among women with smaller and larger waist circumference, stratified by menopausal status.

**Table 1 diagnostics-14-00001-t001:** Demographic, family history, lifestyle, anthropometric, and clinical characteristics in pre- and postmenopausal women with endometrial cancer.

Characteristic	Premenopausal Women		Postmenopausal Women			
**Background**	Median	IQR	Median	IQR	Uncorrected *p*-value	Significance after Benjamini–Yekutieli procedure
*Age*	48	43–52	66	61–72	NA	
*Parity*	*N*	%	*N*	%	<0.001	significant
0	54	34%	230	17%		
1	40	25%	275	20%		
2	52	32%	588	43%		
3+	15	9%	284	21%		
**Family history of cancer**	*N*	%	*N*	%		
Endometrial	7	4%	94	7%	0.311	not significant
Ovarian	1	1%	20	1%	0.717	not significant
Breast	16	10%	149	11%	0.893	not significant
Colon	26	16%	103	7%	0.001	significant
Other	30	19%	278	20%	0.679	not significant
**Lifestyle**						
*Smoking*					0.398	not significant
Never	122	76%	1047	76%		
Former	19	12%	198	14%		
Present	20	12%	132	10%		
*Exercise*					0.243	not significant
Never	53	33%	515	37%		
>20 min, 1–2/w.	46	29%	436	32%		
>20 min, ≥3/w.	44	27%	328	24%		
Strenuous, 1/w.	10	6%	54	4%		
Strenuous, ≥2/w.	8	5%	44	3%		
*Alcohol use*					0.193	not significant
Never	61	38%	596	43%		
0–6 units/w.	89	55%	691	50%		
7–12 units/w.	9	6%	85	6%		
>12 units/w.	2	1%	5	0%		
**Anthropometric**						
*Constitution*					0.024	not significant
Abdominal adiposity	70	43%	742	54%		
Female adiposity	32	20%	259	19%		
Lean	59	37%	376	27%		
	Median	IQR	Median	IQR		
*Weight (kg)*	77	65–93	75	65–89	0.671	not significant
*Height (cm)*	165	162–170	163	159–168	NA	
*BMI (kg/m^2^)*	27.6	23.1–34.6	28.4	24.4–33.2	0.290	not significant
*Waist circumference (cm)*	92	82–108	98	87–112	0.002	significant
*Bra cup size*	*N*	%	*N*	%	0.87	not significant
AA-B	55	34%	443	32%		
C-D	87	54%	756	55%		
E-I	19	12%	178	13%		
**Co-morbidity**						
*Current use of hormonal treatment*	18	11%	160	12%	1	not significant
*Type 2 diabetes*	11	7%	235	17%	NA	
*Hypertension*	35	22%	730	53%	NA	
**Bleeding history**						
*Abnormal uterine bleeding*	140	87%	1226	89%	0.428	not significant
	Median	IQR	Median	IQR		
*Duration of abnormal uterine bleeding (months)*	5	3–12	3	2–6	<0.001	significant

**Table 2 diagnostics-14-00001-t002:** FIGO surgical stage, histology, and risk group in pre- and postmenopausal women with endometrial cancer.

Characteristic	Premenopausal Women		Postmenopausal Women			
	*N*	%	*N*	%	Uncorrected *p*-value	Significance after Benjamini–Yekutieli procedure
**FIGO surgical stage**					<0.001 ϕ	significant
IA	135	84%	801	58%		
IB	9	6%	315	23%		
II	7	4%	79	6%		
III	10	6%	159	12%		
IV	0	0%	23	2%		
**Histologic subtype**						
*Endometrioid*	158	98%	1172	85%	<0.001 Ω	significant
Grade 1	91	58%	512	44%	0.0012 Ψ	significant
Grade 2	49	31%	463	40%		
Grade 3	18	11%	197	17%		
*Non-endometrioid*	3	2%	205	15%	NA	
Serous	0	0%	91	44%		
Carcinosarcoma	2	67%	39	19%		
Clear cell carcinoma	0	0%	33	16%		
Mixed cell carcinoma	1	33%	35	17%		
Undifferentiated	0	0%	7	3%		
**ESMO-ESGO-ESTRO risk group ε**					<0.001	significant
Low	126	78%	636	46%		
Intermediate/High	35	22%	741	54%		

Fisher’s exact test was used for categorical data and Mann–Whitney *U* test for continuous data. ϕ Comparison of FIGO stage IA and ≥IB. Ω Comparison of endometrioid and non-endometrioid. Ψ Rank of grade among endometrioid. ε ESMO risk group: Low (grade 1 + 2, FIGO stage IA), Intermediate/High (all other). Based on hysterectomy if available, otherwise on pre-operative biopsy.

**Table 3 diagnostics-14-00001-t003:** Sonographic findings in pre- and postmenopausal women with endometrial cancer.

Finding	Premenopausal Women		Postmenopausal Women			
**Sonographic findings, all women**	*N*	%	*N*	%	Uncorrected *p*-value	Significance after Benjamini–Yekutieli procedure
**Endometrium**					0.356	not significant
Measurable	154	96%	1274	93%		
Not measurable	4	2%	45	3%		
Not visible	3	2%	58	4%		
**Tumor**					0.001	significant
Defined	118	73%	1157	84%		
Not defined	43	27%	220	16%		
**Myometrium**						
Fibroid present	48	30%	503	37%	0.099	not significant
Adenomyosis suspected	25	16%	96	7%	0.001	significant
**Sonographic findings, in women w. visible endometrium only**						
**Endometrial–myometrial border**					<0.001	significant
Regular	61	39%	298	23%		
**Endometrial echogenicity**						
*Uniform*	65	41%	496	38%	0.387 ϕ	not significant
Hyperechoic	49	75%	377	76%		
Hypo-/iso-/three layered	16	25%	119	24%		
*Non-uniform*	93	59%	823	62%		
Homogenous w. cysts	9	10%	70	9%		
Heterogenous without cysts	77	83%	661	80%		
Heterogenous w. cysts	7	8%	92	11%		
**Bright edge sign**					0.007	not significant
Yes	11	7%	191	14%		
**Endometrial midline**					<0.001	significant
Seen	37	23%	147	11%		
Undefined/not seen	121	77%	1172	89%		
**Color score**					0.545 Ω	not significant
No flow	25	16%	267	20%		
Minimal flow	32	20%	246	19%		
Moderate flow	65	41%	436	33%		
Abundant flow	36	23%	370	28%		
**Vascular pattern**					0.439 Ψ	not significant
No flow	25	16%	267	20%		
Single +/− branching	21	13%	150	11%		
Multiple vessels, focal	26	16%	214	16%		
Multiple vessels, multifocal	57	36%	519	39%		
Scattered vessels	28	18%	168	13%		
Circular	1	1%	1	0%		
**Endometrial thickness (mm)**	14	10–23	17	10–26	0.132	not significant
**Tumor volume (cm^3^)**	7.3	2.4–23.3	7.9	2.7–21.1	0.786 ε	not significant

Fisher’s exact test was used for categorical data and Mann–Whitney *U* test for continuous data. ϕ Comparison of uniform vs. non-uniform. Ω Comparison of no flow + minimal flow vs. moderate flow + abundant flow. Ψ Comparison of multiple vessels, multifocal vs. all others. ε On those with a defined tumor only.

**Table 4 diagnostics-14-00001-t004:** Association of sonographic features and anthropometric factors to ESMO-ESGO-ESTRO risk groups, stratified by menopausal status.

Finding	Premenopausal Women						Postmenopausal Women					
	Low risk		Intermediate/high risk				Low risk		Intermediate/high risk			
**Sonographic findings, all women**	*N*	%	*N*	%	Uncorrected *p*-value	Significance after Benjamini–Yekutieli procedure	*N*	%	*N*	%	Uncorrected *p*-value	Significance after Benjamini–Yekutieli procedure
**Tumor**					0.196	not significant					<0.001	significant
Defined	89	71%	29	83%			509	80%	648	87%		
Not defined	37	29%	6	17%			127	20%	93	93%		
**BMI**					0.696	not significant					<0.001	significant
<30	77	61%	23	66%			338	53%	473	64%		
≥30	49	39%	12	34%			298	47%	268	36%		
**Waist group**					1.000	not significant					0.001	significant
<80 cm	49	39%	13	37%			146	23%	229	31%		
≥80 cm	77	61%	22	63%			490	77%	512	69%		
**Sonographic findings, in women w. visible endometrium only**												
**Endometrial–myometrial border**					<0.001	significant					<0.001	significant
Regular	58	46%	3	9%			207	33%	91	13%		
Non-regular	68	54%	29	91%			415	67%	606	87%		
**Endometrial echogenicity**					0.232	not significant					<0.001	significant
Uniform	55	44%	10	31%			266	43%	230	33%		
Non-uniform	71	56%	22	69%			356	57%	467	67%		
**Endometrial midline**					0.818	not significant					0.431	not significant
Seen	29	23%	8	25%			74	12%	73	10%		
Undefined/not seen	97	77%	24	75%			548	88%	624	90%		
**Color score**					0.156	not significant					<0.001	significant
No flow–minimal flow	49	39%	8	25%			324	52%	189	27%		
Moderate flow–abundant flow	77	61%	24	75%			298	48%	508	73%		
**Vascular pattern**					0.216	not significant					<0.001	significant
All others	84%	67%	17	53%			449	72%	351	50%		
Multiple vessels, multifocal	42	33%	15	47%			173	28%	346	50%		

## Data Availability

The data presented in this study are available on request from the corresponding author. The data are not publicly available due to a lack of ethical approval for the public release of personal research data.

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
