# Peer review of "Sonographic, Demographic, and Clinical Characteristics of Pre- and Postmenopausal Women with Endometrial Cancer; Results from a Post Hoc Analysis of the IETA4 (International Endometrial Tumor Analysis) Multicenter Cohort"

_diagnostics, 2023, doi:10.3390/diagnostics14010001_

Round 1
Reviewer 1 Report
Comments and Suggestions for Authors
I could recommend to add concrete numbers to results in abstract. Plenty of readers start reading with summary and they have not enough patience to find it in main text.
Additional Comments:
1. What is the main question addressed by the research?
Reviewer: I only recommend shift data to abstract.
2. Do you consider the topic original or relevant in the field? Does it address a specific gap in the field?
Reviewer: Topic is relevant but it is only confirmation of known information.
3. What does it add to the subject area compared with other published material?
Reviewer: Large of sample.
4. What specific improvements should the authors consider regarding the methodology? What further controls should be considered?
Reviewer: Nothing
5. Are the conclusions consistent with the evidence and arguments presented and do they address the main question posed?
Reviewer: Yes
6. Are the references appropriate?
Reviewer: Yes
Author Response
Your suggestion of adding concrete numbers of the results presented in the abstract has been accepted and changed accordingly.
---------------------------------------------------
Reviewer 2 Report
Comments and Suggestions for Authors
This is an observation cohort study comparing the sonographic, anthropometric, and clinical features between pre and postmenopausal women with confirmed endometrial carcinoma. I have several comments to offer to the authors:
1. It is important to identify features of intermediate/high risk disease in pre and postmenopausal women to allow for adequate counselling with regards to prognosis and hence also to guide appropriate treatment. In this regard, I would consider the findings generated to be of some clinical importance.
2. The findings from this study are still to be considered exploratory given the study design chosen and the authors have, quite rightly, acknowledged this issue in the manuscript.
3. My main concern with the results is the lack of controlling for possible confounding in exploring for possible associations between the menopausal status as well as the risk categories with the independent variables. I appreciate that there may be insufficient power in the study to undertake this type of analysis but it should at least be acknowledged as a limitation of the study in the manuscript.
4. As mentioned by the authors, there is also the risk of multiple statistical testing which is mitigated by the use of the Benjamin-Yekutieli procedure. I am not an expert on this method of statistical correction and it may be wise to seek a statistical opinion on the use of this test.
5. It would be information for the authors to report on how recruitment into this study was performed i.e. were participants recruited consecutively at these 17 gynecological oncology centres? Consecutive enrollment is not necessarily the same as consecutive recruitment.
Author Response
Thank you for your comments. Issue #3, #4 and #5 have been changed in accordance with your recommendations.
---------------------------------------------------
Reviewer 3 Report
Comments and Suggestions for Authors
This properly conducted investigation revealed some insights for the stratification of menopausal status regarding the presentation of EC, and the associated sonographic findings. However, the identification of the waist size difference between (presumably younger) pre-menopausal and post-menopausal women should be expected.
This can be a valid contribution to the medical literature.
Author Response
Thank you for your comments. As I cannot see any specific suggestions for improvement of our manuscript based on your comments, I haven't made any changes.